# Toward an Intelligent Blockchain IoT-Enabled Fish Supply Chain: A Review and Conceptual Framework

**DOI:** 10.3390/s23115136

**Published:** 2023-05-28

**Authors:** Shereen Ismail, Hassan Reza, Khouloud Salameh, Hossein Kashani Zadeh, Fartash Vasefi

**Affiliations:** 1School of Electrical Engineering and Computer Science, University of North Dakota, Grand Forks, ND 58202, USA; 2Department of Computer Science and Engineering, American University of Ras Al Khaimah, Ras Al Khaimah 72603, United Arab Emirates; 3Department of Mechanical Engineering, University of North Dakota, Grand Forks, ND 58202, USA; 4SafetySpect Inc., 10100 Santa Monica Blvd., Suite 300, Los Angeles, CA 90067, USA

**Keywords:** Blockchain, fish supply chain, Internet of Things, machine learning, traceability, security, anti-fraud

## Abstract

The fish industry experiences substantial illegal, unreported, and unregulated (IUU) activities within traditional supply chain systems. Blockchain technology and the Internet of Things (IoT) are expected to transform the fish supply chain (SC) by incorporating distributed ledger technology (DLT) to build trustworthy, transparent, decentralized traceability systems that promote secure data sharing and employ IUU prevention and detection methods. We have reviewed current research efforts directed toward incorporating Blockchain in fish SC systems. We have discussed traceability in both traditional and smart SC systems that make use of Blockchain and IoT technologies. We demonstrated the key design considerations in terms of traceability in addition to a quality model to consider when designing smart Blockchain-based SC systems. In addition, we proposed an Intelligent Blockchain IoT-enabled fish SC framework that uses DLT for the trackability and traceability of fish products throughout harvesting, processing, packaging, shipping, and distribution to final delivery. More precisely, the proposed framework should be able to provide valuable and timely information that can be used to track and trace the fish product and verify its authenticity throughout the chain. Unlike other work, we have investigated the benefits of integrating machine learning (ML) into Blockchain IoT-enabled SC systems, focusing the discussion on the role of ML in fish quality, freshness assessment and fraud detection.

## 1. Introduction

Fish products have long been traded domestically and internationally. Raw fish often undergo several processing stages before they reach their final destination; however, they may be subjected to illegal, unreported, and unregulated (IUU) activities during this journey, which could be hundreds of kilometers. Examples of fish/seafood IUU activities may include illegal fishing, mislabeling, counterfeiting, and substitution fraud [1,2,3]. Consumers, retailers, and the global seafood industry’s reputation are adversely affected by fish fraud such as in the case of substituting fish species with cheaper varieties [4]. The “Illegal, Unreported, and Unregulated Fishing Enforcement Act” aims to combat IUU fishing and prevent illegally harvested fish from entering the ports and markets. IUU fishing robs nations of up to USD 23.5 billion annually [5]. The seafood industry is highly vulnerable to fraud due to factors such as the similar appearance of many species, price variations, complexities in SCs, and challenges with supply and demand [6].

Instances of seafood fraud are frequently reported; however, many incidents go undetected, and the full extent of seafood fraud is difficult to determine. Existing rapid DNA testing can only detect a single species with each assay, which mandates new technologies to detect mislabeling rapidly. Detecting seafood mislabeling requires innovative approaches that can measure the compositional and chemical characteristics of seafood. Portable and cost-effective IoT-enabled inspection tools that use efficient machine learning (ML) techniques are needed to assess seafood fraud more comprehensively and to mitigate its potential impacts. Estimating the apparent consumption of mislabeled products is currently limited by the quality of the data on consumption and mislabeling. Mislabeling rates can be integrated with import and production data to produce estimates of apparent mislabeled seafood consumption. The United States seafood industry faces unprecedented challenges as consumer preferences change and market access is threatened. There are two problems related to species identification that are costly for consumers, retailers, and suppliers: (i) illegally harvested seafood sold to the market and (ii) labeling legally caught seafood as other species with higher value. There is an unmet need for identifying seafood species; they must be affordable, fast, accurate, and portable enough to be used at any stage in seafood SC. In addition, growing domestic and export demand by strengthening the US seafood brand is vital for preserving the fish and seafood industry. Traditional SC can go through numerous chain parties and span multiple domestic and international intermediaries, which makes it difficult to pinpoint real-time track and trace data to ensure the product’s safety or quality. As a result, technological advancements that digitally document the seafood’s origin, continuously monitor the product’s details, and record stages of SC life cycle are essential to ensure consumer confidence in the product’s source, safety, and quality and to allow government food authorities and global agencies to effectively ensure regulatory compliance with food standards.

Separate SC participants have maintained information in local databases or have used traditional bookkeeping methods, which make it difficult to track and trace fish products from source to destination. These simple SC methods do not prevent illegal and unethical access to data; hence, they result in an increased need for effective IUU prevention and detection methods to build an entirely trustworthy and fully traceable SC system. This system aims to track and trace fish products and to record the GPS coordinates throughout the path from where it was caught until it reaches its final destination.

Blockchain is one recent technology promoted to support perishable products’ traceability and authenticity, especially when integrated with the Internet of Things (IoT) [7]. In addition to food SC, other vital applications progress and develop in terms of reliability, availability, and security including healthcare, e-governance, and the automotive industry [8,9,10]. The authors of [11] believe that the global SC will be one of the most promising trends for integrating Blockchain and IoT by 2030. Blockchain with IoT integration should enhance secure data collection, allow for long-term preservation, and prevent IUU practices [12], creating an entirely trustworthy and fully traceable SC system.

IoT-enabled devices, such as wireless sensors, RFID, GPS chips, and spectroscopy imaging devices, are deployed to sense, actuate, and collect data observations. These data are then securely registered on a distributed unchangeable database system available for stakeholder access [13]. Combining other emerging technologies, such as big data, edge computing, and artificial intelligence allows stakeholders to realize additional significant benefits of Blockchain [14], creating smart SC systems in terms of enabling product trackability and traceability, efficiently controlling product safety and quality, and promoting prevention and detection methods against IUU activities.

Traceability is a fundamental feature of a Blockchain-based SC system that enables stakeholders to trace the fish product back to the point of harvest [15]. Tracing occurs while the product navigates the processing, manufacturing, distribution, and shipping stages up to delivery to end consumer, allowing stakeholders to keep track of the biological and physical characteristics of the product as well as its surrounding, such as temperature, humidity, and microbiological features. The Blockchain adoption rate in SC management and logistics is still low since stakeholders have yet to realize the benefits of Blockchain to the fish industry and to tangibly measure its value; therefore, they are not fully aware of its potential to improve the trackability and traceability of fish and seafood products.

The primary contribution of this paper is to review the research efforts that have been directed toward having smart Blockchain IoT-enabled fish SC systems. We discuss traceability in both traditional and smart SC systems that employ Blockchain and IoT technologies. We also examine the key design considerations in terms of traceability and other key quality attributes to consider while designing smart Blockchain-based SC systems. In addition, we propose a layered architecture for an intelligent Blockchain IoT-enabled fish SC framework to share the data and validated status of the fish products throughout harvesting, processing, packaging, shipping, distribution, and final delivery. We aim to assist researchers toward integrating ML, Blockchain, and IoT technologies in designing fish SC systems that verify product authenticity and quality and promote effective methods to detect and prevent IUU activities, such as fish fraud and mislabeling.

This work is organized as follows: Section 2 reviews the literature for the applicability of Blockchain in fish SC from industry and academic perspectives. Section 3 defines traceability in both traditional and modern SC. In addition, this section illustrates the key quality attributes and their associated architectural design considerations when designing Blockchain-based SC systems. Section 4 presents our proposed Blockchain IoT-enabled fish SC in terms of system requirements and architectural framework in detail. Section 5 discusses ML integration with Blockchain IoT-enabled SC in terms of related work and goals of integration, followed by a commercial use case demonstration. Section 6 concludes the paper with key challenges and future work.

## 2. Literature Review

### 2.1. Industry Perspective

The food industry has been disrupted by the COVID-19 pandemic, which has lead to bottlenecks in the SC due to raw material unavailability, price inflation, production fluctuation, labor shortage, etc. [16]. In addition, food fraud has increased due to the COVID-19 pandemic, estimated to cost the food industry USD 49 billion annually. Fish and seafood as a labor-intensive industry face the same challenges, such as labor shortage, logistics barriers, and changes in consumer buying power [17]. To mitigate the post-pandemic disruptions, many countries and companies are working to eliminate food fraud though the development of trustworthy, secure, transparent, and resilient food SC systems.

The global industry for seafood was estimated at USD 113.2 billion in 2020 and is expected to reach USD 138.7 billion by 2027, according to the Seafood Global Market report. At least 90% of all seafood comes from developing nations. The Marine Stewardship Council conducted a consumer survey at the end of 2020, revealing that 56% of consumers are willing to pay more for seafood from certified sustainable fisheries [18]. The SC has moved toward using emerging technologies, such as Blockchain, IoT, and ML. Various SC functions can benefit from this integration, such as logistics, inventory management, and quality control. Developing transparent decentralized traceability SC systems are able to provide economic and operational benefits compared to conventional SC, such as reducing operational costs and making correct and timely decisions while assuring data immutability and public accessibility [19]. Such smart SC systems help the industry operate more efficiently and profitably.

Fish SC systems can be defined as a dynamic group of interactions between harvesters, suppliers, public health authorities, manufacturers, shippers, retailers, and customers. Since fish and seafood are vital temperature-sensitive products and due to the apparent increase in IUU practices in the seafood industry, governments and stakeholders are demanding to know the stories behind their seafood, motivated by the consumers’ awareness of quality and safety requirements and global concerns about having certified seafood from sustainable sources. Therefore, traceable SC systems aim to deliver the right product at the right price, in the right condition and quality, and at the right time [20].

Blockchain-based industry solutions for seafood SC have been launched commercially [21] and developed by large technology corporations and startups such as IBM, SAP, Oracle, and Microsoft [22]. Provenance, one of the first Blockchain-based solutions introduced in 2013, enables consumers to trace fish products back to the fisherman. Fishcoin, established in 2018, is another Blockchain-based seafood traceability system developed to establish trusted data sharing between stakeholders [23]. IBM Food Trust is a food SC solution built on Blockchain technology that was also started in 2018. Big retailers, including the Sustainable Shrimp Partnership, Nestle, Walmart, Carrefour, and Unilever, have joined the IBM food trust.

Bumble Bee Foods and SAP developed a Blockchain-based system in 2019 to track and trace fresh fish from the source to the end consumer. OpenSC [22] is an online Blockchain platform launched in 2019 and founded by WWF Australia and BCG Digital Ventures. OpesSC is integrated with IoT-enabled devices and ML classification techniques and can track fish from the source and along its journey in the SC.

Tracey is a Blockchain-enabled application developed to trace seafood products in the SC. This application, in cooperation with UnionBank from the Philippines, allows fishers to relate to their marketplace and access an easy, bankable solution. TraSEAble is a Blockchain-based system for seafood and agriculture products that has traceability integrated with IoT to ease stakeholders’ collaboration. Pacifical, in cooperation with Atato in 2018 [24], launched a Blockchain-based traceability solution through a public Ethereum platform to ease collaboration between stakeholders. Fiji is a pilot project reported in [1] that uses Blockchain to build a transparent and traceable tuna SC using Ethereum to prevent illegal and unsustainable fishing practices in the Pacific Islands tuna industry.

### 2.2. Academic Perspective

Figure 1 depicts the three IoT, Blockchain, and SC integration levels: (1) the level of Blockchain, IoT, and SC; (2) the specific level of Blockchain, IoT, and the food supply chain; (3) and the level most relevant to this work: Blockchain, IoT, and the fish supply chain. This section focuses on reviewing recent SC works that have addressed the tracking and tracing of fish and seafood products with the adoption of Blockchain technology.

The authors of [25] considered the impact of using Blockchain integrated with IoT in SC systems by examining several case studies. Among the case studies covered in this paper is a pilot project that analyzed the fish SC in Indonesia since it is the world’s largest tuna producer. The project’s goal was to track the fish sources for transparency. The authors of [21] developed a Blockchain-based fish farm platform to protect agriculture data from tampering using Hyperledger Fabric. The authors of [26] presented a business case study for implementing Blockchain technology in the Thai fish industry’s SC system.

The authors of [4] reviewed the methods used in the literature to build seafood SC systems. This work focused on improving the enforcement of regulations governing seafood species provenance. Among these technologies is Blockchain, presented as a tamper-proof shared database that can be accessed by regulating authorities and industry partners. The authors of [13] depicted the general idea of integrating Blockchain with IoT-enabled devices for product traceability in the fish SC. The authors of [27] investigated adopting agnostic Blockchain architecture in the fish SC to support the integration of different Blockchain platforms within the same solution.

The authors of [28] developed a Blockchain-enabled system with the Sydney Fish Market (SFM). The system offers trusted fish provenance, real-time SC tracking, and automated quality control. Several experiments have been conducted with SFM and a local SC to validate the system’s feasibility and effectiveness. The project won a prestigious Australian Award in 2020, establishing its ability to enhance consumer confidence and increase sustainable fishing practices. The authors of [29] provided a Blockchain SC model for the Indonesian fishing industry. The paper illustrated how Blockchain could be used in a fishery SC when combined with emerging technologies such as AI, IoT, and big data. The paper also discussed the economic benefits of using Blockchain to reduce costs and to increase production efficiency.

The authors of [30] chose tilapia, a widely traded fish in Ghana, for a case study to illustrate the potential benefits of using Blockchain technology for SC management and logistics. The paper emphasized the importance of using a Blockchain-based system to enhance accessibility, traceability, and communications. The authors also assessed Blockchain adoption in the Tilapia industry, including how this technology could reshape and resolve conventional SC issues.

The authors of [31] investigated the motivation across supplier, transporter, retailer, and final customer toward the adoption of Blockchain in frozen fish SC. The study came to the conclusion that traceability matters for the four types of SC participants and there is a higher likelihood of purchasing fish products that have traceable information available.

## 3. Blockchain Architectural Design Considerations for Traceability in Supply Chain

Blockchain can be defined as a distributed ledger technology (DLT) that is immutable, replicated, and synchronized among a set of untrusted parties in a P2P architecture [20,32,33]. Any modification to the ledger is achieved through consensus between a group of nodes called miners, voters, or peers. Miners are responsible for verifying and validating transactions in terms of data and identity using the selected consensus protocol. Blockchain technology is commonly used in digital cryptocurrency; however, other domains have integrated this technology with IoT to propose secure solutions against data vulnerabilities [3,34]. Blockchain integrated with IoT is used to have efficient mechanisms for node identification, secure authentication and communication, safe resources accessibility, and distributed storage [35,36]. Blockchain technology combined with IoT has various applications in various fields such as SC, agriculture [11], healthcare [37], digital energy and smart grids [38], the automotive industry [39], smart homes [26], and financial applications.

Traceability is an essential quality attribute of SC systems; it delivers knowledge of the product’s movements to system users throughout the product’s lifecycle in the form of track and trace data. This section discusses traceability in traditional data repository systems vs. smart SC systems. We then focus the discussion on Blockchain-based SC systems by examining traceability in addition to other core quality attributes along with their design considerations.

Traceability in SC systems requires: (1) increasing SC visibility, openness, transparency, neutrality, and reliability; (2) improving quality control and regulatory compliance; (3) reducing possible risks and enhancing public safety; and (4) managing logistic operations and enhancing inventory tracking and product recalls. Alternative definitions of traceability in the SC include the ability to trace the data history, such as the chain of custody, associated with the product from the consumer back to the producer [40].

Traceability is architecturally challenging because of a lack of literature on classifying, characterizing, or evaluating a system at the highest level of abstraction with regard to the traceability requirement. There are no current systematic surveys designed to document the body of information on traceability as a quality attribute in the SC system, nor are there any general or domain-specific architectural solutions available to the best of our knowledge.

### 3.1. Traceability in Traditional Supply Chain Systems

Traditional database systems support traceability by collecting data into a planning data repository. These systems have been used to store the traceability data generated during SC operations in a centralized data repository to implement automated digital traceability. Numerous studies have discussed the limitations and constraints of adopting traceability in the SC industry [41] using conventional databases. Examples of these limitations include issues related to logistics, knowledge gaps, and technology limitations. Another study [42] focused on conventional technologies used to track and trace food in SC systems, such as RFID, QR codes and barcodes, DNA-based biosensors, and wireless sensor networks (WSNs). RFID and barcodes are the most popular technologies used in the current SC systems.

The key challenges with these conventional technologies are: (1) the need for line-of-sight communication without obstruction for scanning, (2) difficulties with reading damaged labels, (3) difficulties with collecting environmental information, (4) limited memory size, and (5) lack of interactions and poor communication between the system devices.

### 3.2. Traceability in Smart Supply Chain Systems

Modern SC systems adopt Blockchain in addition to other advanced technologies to enhance traceability. For example, anti-fraud and tracking technologies, such as RFID and WSN [43], enable the system to track the product movement and detect counterfeit and fake products [7]. Smart functioning sensors are used to classify and analyze product quality, such as optical sensing devices, in the form of spectroscopy and imaging instead of conventional devices [44]. Other emerging tracking technologies include integrating ML and advanced computational methods, such as chemometrics, to identify product originality and authenticity [45]. One technical issue is the interoperability between these technologies and database platforms that requires new interoperability standards for the governance of IoT devices and data communication and storage.

### 3.3. Blockchain-Based Supply Chain Design Considerations

Besides traceability, any SC system must possess a quality model that consists of several key quality attributes, including performance, security, privacy, scalability, latency, integrity, usability, and interoperability. These are vital requirements that must be emphasized when designing well-engineered SC systems. Table 1 summarizes the Blockchain-based SC system’s key architectural quality attributes and design considerations. Among the key considerations for each quality attribute are Blockchain type, which could be public, private, or consortium; data storage and computation type, which could be on-chain, off-chain, or hybrid [24,46]; the choice of consensus protocol; and block size and data structure used to record the values or transactions.

Blockchain creates new forms of distributed and decentralized software architectures in which a ledger’s shared state can be reached across a network of untrusted participants without relying on a single authority [47]. The critical technical concerns regarding Blockchain P2P architecture include (de)centralization, storage, computation, network size, and a Blockchain infrastructure setup that includes block size, data structure, consensus protocol, processing speed, transactional rate or transactions per second [24,46].

From a software architecture perspective, Blockchain cannot meet the requirements for all usage scenarios by itself. It is vital to examine the various Blockchain configurations and to assess their impacts on the most important quality attributes to select the best software architecture for an optimal design. We will concentrate on the essential three considerations that are appealing for Blockchain-based SC system design: Blockchain type, data storage and computation, and consensus protocols. We will then assess their impacts on the quality attributes for the overall systems.
Blockchain Type

Blockchain (de)centralization is determined by the access it offers to system entities, either public or limited to certain participants who have the right to access the Blockchain network. Blockchain can be either: (1) fully decentralized (public), (2) fully centralized (private), or (3) partially centralized or partially decentralized (consortium). Public Blockchains have better transparency, audibility, and scalability; however, they sacrifice privacy. Private Blockchain is typically more efficient and has fewer nodes than public Blockchain, but it usually sacrifices decentralization for speed. Consortium Blockchains are controlled by a set of nodes with privileges to validate transactions communicated among multiple organizations. Private or consortium Blockchains offer better options for SC systems since the network is usually governed and hosted by one or few organizations; thus, they control the number of participants and protect their sensitive information by specifying through ACL and smart contracts.
Data Storage and Computation

Data storage and computation operations can be performed on-chain or off-chain, or with external storage. The heavyweight approach involves storing data and metadata (or the state of data) on-chain. The lightweight approach involves storing the essential data and metadata on-chain at a core ledger, allowing it to be accessed by the public, while the actual data and the data required by smart contracts for verification and documentation are stored off-chain [48]. Off-chain data are usually linked to on-chain data through a type of hash identification generated using hashing algorithms such as the Secure Hash Algorithm (SHA-256). Off-chain storage is useful for keeping large files that contain additional information or when storing information requires more space than the Blockchain supports [33,49]. Hybrid Blockchain on-chain and off-chain storage models in SC systems have been proposed in the literature [50,51] to balance the storage load while preserving efficient data retrieval. The InterPlanetary File System (IPFS) is an example of a distributed file storage system for off-chain implementation [52].
Consensus Protocols

Consensus protocols are responsible for maintaining integrity and security in Blockchain-based systems [53]. Several consensus protocols have been used in the literature to enforce a majority agreement between nodes to validate transactions. Every new block of verified transactions must be appended to the existing chain under the agreement of the consensus protocol. Examples of these protocols include Proof of Work (PoW), Proof of Sake (PoS), Proof of Authority (PoA), Proof of Elapsed Time (PoET), and Proof of Byzantine Fault Tolerance (PBFT) [20]. Each protocol has its own set of benefits and drawbacks, and each is best suited for a specific application. Lightweight consensus is preferable for SC [54]. Several solutions have been proposed for SC systems based on protocols such as PoA [53], PBFT [55], and new versions of PBFT. The consensus in private or consortium Blockchains is likely to be faster than the consensus in public Blockchains since the transaction validations in private or consortium Blockchains are performed by certain stakeholders, while transaction validation is performed by a majority vote in public Blockchains [56].

## 4. Proposed Blockchain IoT-Enabled Fish Supply Chain

### 4.1. System Requirements

System requirements are represented by identifying system use and abuse or misuse cases. Use cases generally describe the list of actions or event steps that define the interactions between an actor or a device and a system to achieve a specific function. Understanding the proposed system’s use cases for each actor or device can help identify the functionality of each entity in the system and how it responds to requests. Abuse or misuse cases define the acts that cause intentional system violations; therefore, security requirements must be determined. The system is exposed to one or more of the following events under these circumstances, including (1) faults and failures in data generation or device functionalities or (2) cyber-attack target system devices or misbehaving actors aimed at risk system security, which can be identified by abuse or misuse cases. Examples of abuse or misuse cases include product fraud, sensor tampering, sensor feed modification, network cyber-attacks, and transaction repudiation. The proposed Blockchain-based fish SC is a complex dynamic system that can be defined through (1) a set of use cases describing the transactional processes on the SC for fish traceability back to its origin and (2) a set of abuse cases that determine the security requirements needed to control IUU activities, such as fraud, substitution, and possible risks, at any point in the chain. Table 2 and Table 3 present examples of a use case description for “reading observation from an IoT-enabled sensor” and a misuse case description of “fraud through product data tampering”, respectively.

### 4.2. Architectural Framework

Our proposed Blockchain IoT-enabled fish SC framework is designed using layered architectural style (Figure 2). The proposed framework consists of four architectural layers: SC, IoT, knowledge, and application. The SC layer depicts the interactions between the actors who play the role of nodes that are communicating and transacting with each other in the chain’s physical flow. The IoT layer is presented by the IoT-enabled devices, sensors, and actuators that are connected through the IoT cloud to the SC system to monitor and collect traceability data and to be stored in the Blockchain. The knowledge layer, or Blockchain layer, stores all data activities, or transactions, that occur during the SC operations in the form of data blocks that are encrypted and controlled by the smart contracts and distributed to each involved entity. The application layer primarily provides the applications that extract the different functionalities and then integrates the end-users with system services so that they can access the data.

#### 4.2.1. Supply Chain Layer

The SC layer represents the physical flow of the fish product, which begins with harvesters or suppliers who capture or harvest fish species as raw materials from their natural sources such as sea, lakes, rivers, or aquaculture farms. Fresh fish are then typically bid and sold at fishery auction traders and packaged and sent through international and domestic intermediaries, such as processing plants, manufacturers, distributors, and retailers, and then to the end customers.

Fish can be sold, processed, packaged, stored, and transported multiple times by different intermediaries during their life cycle in the SC [13]. For example, fish may be frozen, salted, or transformed into several types of fish products, repackaged in bulk or individually, and then shipped to distributors to be stored or distributed again to retailers. Mass products might be repackaged into individual items to be displayed for end consumers at retail stores and markets.

Fish is a temperature-sensitive product that requires the upkeep of cold chain throughout its journey, from the fishery to consumer. To comply with the regulations in terms of temperature and other conditions, IoT-enabled vehicles or refrigeration trucks are used for distribution, which are four to eight times more expensive than normal logistic services [25]. These refrigeration trucks keep the fish product safe in a controlled environment; therefore, every stage of the SC is typically followed by a quality and safety assessment to ensure compliance with regulations. The SC entities can be divided into the following primary stakeholders associated with their major activities:
1.Harvester (raw fish supplier): fish catching, registering, and packaging.2.Manufacturer (factories, fishing docks): grading, fish processing, manufacturing, registering products each with a unique number, and packaging. Fish products must be re-registered and identified with a unique batch number if they are repackaged into batches.3.Distributor (delivery companies, warehouses, storage hubs): packaging, delivering, classifying, quality checking, standardizing, tracking, and storing.4.Safety and quality regulators (government food safety inspectors, certifiers, auditors): inspecting, grading, penalizing, licensing, and standardizing. Inspectors, certifiers, and auditors can also be classified as system users responsible for inspection, auditing, test reporting, and issuing product certifications [57].5.Retailers (markets, supermarkets, wholesale stores, retail shops): receiving, packaging, classifying, selling, storing, distributing, and marketing.6.Customers: buying, returning, quality checking, reporting, and consuming. Customers or end-consumers usually interact with the system interface by querying fish product data, which are permanently and securely stored in the Blockchain. Blockchain-based SC systems do not usually classify the consumer as a stakeholder since they consume the product at the end of its lifecycle; therefore, they do not need an account on the Blockchain.7.Other actors may involve software developers and project coordinators.

The actors in the proposed Blockchain IoT-based SC system can be divided into three categories: non-authenticated parties, authenticated parties, and anonymous. Non-authenticated parties represent parties that have not yet been authenticated themselves. Those parties are required to register an account, which will be accepted or denied by the system. Authenticated parties represent those who are permitted to join the system. The platform provides different access credentials for each party, allowing them to perform specific functions such as add, edit, remove items, or flag them as read-only. Anonymous users do not need to register or be authenticated and can only access the system to consult system information and query product data.

#### 4.2.2. IoT LAYER

The IoT layer can be imagined as a three-tier architecture consisting of IoT-enabled devices, edge servers, and the cloud (Figure 3). At this layer, Blockchain supports data security and trust establishment, while edge servers move the computation load and storage at the edge devices instead of being completely performed at the cloud, ensuring higher system efficiency and scalability [58,59]. The distributed architecture of the IoT layer can integrate computation, storage, networking, and application processing to provide better resources to end users.

Edge computing can better address issues, such as latency, limited power, limited bandwidth, security, and privacy [60], than its opposite technology, cloud computing. For example, edge computing allows for better use of local/cloud resources, minimizes the response time, and reduces power consumption by moving storage and computation from the cloud to edge devices. The proposed system involves deploying Blockchain on the cloud for time and energy-consuming data mining processes with large data volumes, while data sensing and preprocessing for small amounts of data are performed locally at the edge devices. This process is made possible by deploying smart functioning sensors that incorporate microelectromechanical system (MEMS) technology [61]. These sensors can perform data processing and analysis at or near the source, reducing data movement between the platform and the device. Intensive computing tasks can be offloaded from IoT devices to edge servers up to the cloud [14] for better load distribution.

The devices of the IoT-enabled devices tier have scarce resources; therefore, their participation in the Blockchain network as data sources is facilitated by more capable servers found on the upper layers, at the edge, and on the cloud. The edge server tier is located in the middle of the architecture near the IoT-enabled devices (Figure 3). Edge servers can efficiently perform computational and analytical tasks, including hash computation, encryption and decryption, and mining. These tasks are offloaded from the devices and are outsourced to edge servers for execution and data analysis. Edge technology, equipped with analytical models in IoT settings, can be used to securely perform diagnostic, descriptive, and predictive analytics faster and cheaper. The cloud tier, located on top of the network architecture, consists of interconnected cloud servers that represent a large data center that provides core cloud services and process requests from the edge servers [60].

IoT devices, including RFID devices, machine-readable optical labels, smart weighing devices, survey cameras, handheld contamination and inspection devices, and GPS trackers in addition to wireless sensors, are responsible for collecting the track-and-trace data in a timely manner. Such devices monitor physical and chemical parameters such as temperature, humidity, dissolved oxygen, salinity, and carbon dioxide [62]. The proposed system has the following IoT-enabled devices:1.Wireless sensors are usually placed at strategic points on the chain and have identification and sensing capabilities. These sensors communicate precise measurements continuously, called polling, or upon request. For example, a truck’s temperature monitoring sensors continuously measure cargo hold temperature and report the readings to the system. Each reading, in the form of a data transaction, must contain a unique track-and-trace number for the current measurement associated with the sensor device’s physical address [63]. The sensor keeps the observations in the sensors’ memory to be communicated with the server or can be automated to be reported regularly. Smart contracts can be utilized to control and regulate the sensors and to trigger the system if the readings are out of the specified limits.2.IoT-enabled optical scanning devices can read RFID tags or machine-readable optical labels such as QR codes, and barcodes report product information with the track-and-trace readings. For example, RFID tags are unique digital cryptographic identifiers that connect physical items to their virtual identities [64], typically attached to the fish containers or packing cases and programmed to log trace data. Barcodes are usually used to label individual products; however, RFID is more convenient than barcodes but has a higher cost.3.Smart weighing devices are used to weigh the fish caught during fishing operations. Weight logging could be automated to forecast the time between the landing date and the selected destination.4.On-board survey cameras and electronic monitoring systems can identify interactions with by-catches and protected fish species.5.GPS trackers can be used for real-time location determination and detailed tracking information, including geo-location, speed, and time.6.Automated handheld imaging inspection devices can check fish freshness and possible microbiological and chemical contamination in fish species or fish farms. For example, the Adulteration and Traceability (QAT) handheld device is a proprietary technology proposed and developed by SafetySpect [65,66] that can be used to measure fish freshness and fecal contamination. This device can also be used to inspect several other types of meat products. The QAT device is portable, easy to use, and efficiently detect possible contamination.

#### 4.2.3. Knowledge Layer

The knowledge, or Blockchain, layer can be conceptualized using proper Blockchain ontology, which is related to the the potential enterprise adopters’ business operations and processes [67]. Blockchain ontology is crucial for understanding how data are exchanged over the Blockchain-based network [41] since it provides the system framework with a structured knowledge representation [68] that helps to distinguish the different key components, including nodes, transactions, blocks, smart contracts, ledgers, and consensus (Figure 4). Blockchain ontology is responsible for describing and addressing components such as sensors, actuators, objects, devices, and services and their relationships while providing an essential level of abstraction to manage heterogeneity and interoperability.

The nodes are interconnected in a P2P distributed Blockchain network. Exchanged transactions have information about product movement as the product moves through the SC (Figure 5). Transactions include data about events related to transportation, processing, testing, value-added activities, packing, storage, and logistics actions. The transactions contain geo-locations, departure and arrival times, transit times, and operations such as processing, sorting, packaging, quality checking, farm names, sensor readings, fish types, batch numbers, batch quantities, best-by dates, expiration dates, brands, colors, product labels, weight, storage conditions, inspections, and quarantine information. SC management consists of different types of transactions in which each transaction has specific data fields associated with each particular entity or object. The number of transactions and their execution times are expected to be reduced for all transactions with the trust established using Blockchain. Transactions are verified, validated, and bundled into blocks once agreed upon by peer nodes and then are securely stored on the distributed ledger to be accessed only by stakeholders.

The distributed digital ledger is a collection of replicated, shared, and synchronized data spread across registered and authorized Blockchain network participants. The ledger provides secure temper-proof storage for IoT physical device configurations and sensing data. Peers vote on any update to agree upon a change if modifications are made to the ledger. Each node then receives a copy of any changes recorded on the ledger. This process makes the system more transparent, reliable, and, most importantly, without third-party interference [69].

Smart contracts are another key Blockchain component that ensure the system’s transparency, security, and autonomy. Smart contracts use the agreed-upon trade rules and regulations to ensure trust between stakeholders joining the Blockchain network. Smart contracts also add flexibility and power to program the SC system’s business logic aligned with preset conditions in cases such as order verification, inventory update, and payment trigger. This way, requirements and compliance with regulations are ensured, and traceability constraints are enforced. Smart contracts are automatically triggered when a set of conditions are met. For example, manufacturers and processing companies can bid on fish from fisheries using a smart contract and can set the order when specific parameters are met, allowing the fish product to move smoothly through the SC.

Data contained in the Blockchain can be categorized into track, trace, and informative. Track data record a history of the product’s microbiological and physicochemical parameters readings, such as temperature, humidity, and microbiological information, which are used to recognize any changes or variations in the ingredients or conditions. Trace data are used as to create a graph of coordinates on a geographical map that can be used to represent the past and current location history. This information is necessary to ensure the fish product’s safety and quality at each stage of the chain. Informative data are the nutrition information that supplements track-and-trace information. These data are used to enhance the product’s end stories presented to the customer. The system presents the track-and-trace data in the form of a chain of events when stakeholders try to retrieve the data associated with a fish product.

The need to store massive amounts of data is one of the key challenges that causes a significant reduction in system performance. One of the solutions proposed in the literature is using a hybrid Blockchain storage model, where on-chain stores the primary ledger and metadata, while off-chain stores the large amounts of data required by smart contracts for verification and documentation [13]. An example of using off-chain is when a retailer purchases a fish product from a supplier. The retailer determines the product quantity with the required specifications to the supplier, who then delivers the products with the appropriate specifications through a selected shipper. The shipper delivers the product to the retailer and documents the delivery. The purchase and delivery agreements are stored off-chain with on-chain hash evidence. The shipper’s merchandise delivery evidence should also be stored off-chain. Cryptographic hashes and meta data are kept on the ledger to identify the corresponding data off-chain.

Private Blockchain can be the best option for applications that have sensitive business data or that place restrictions on who can participate in the network and who has the authority to access the data [42]. ACLs add granular data access rights based on each participant’s role in the system [44] and manage trust through rewards or penalties based on the smart contract’s output [47]. Hyperledger Fabric can be used to implement the proposed Blockchain’s database system where all participating nodes’ identities are known and authenticated and only a pre-selected set of nodes can validate and verify transactions. Hyperledger Fabric is a private, open-source Blockchain platform used for enterprise solutions [70] that can be deployed at the enterprise level for sensitive data sharing and exchange in a trade context [71]. Hyperledger Fabric properly ensures security, interoperability, and privacy [46]. The platform allows the system to configure some basic features, such as the number and size of transactions in a block, which impact the network throughput and latency. The transaction separation flows into three steps: execution or endorsement, ordering, and validation. This separation is the primary distinction between Hyperledger Fabric and other platforms [48], and it assists with scaling the solution in terms of the number of processing transactions and the number of participating nodes.

Permission to read, write, and validate transactions to the ledger is chosen by consensus. Access to the entire ledger is not uniform across all stages in the SC to avoid misusing sensitive business information; however, the transacting nodes have access to the complete history of transactions, including product location and quality information. This way, fish product traceability is maintained, and safety and quality violations or fraud can be identified. Hyperledger Fabric enables a clear line of communication between multiple players in the SC while preserving confidentiality since it uses the concept of channels, which allows the fabric network to be partitioned into multiple Blockchains. For example, separate channels can be created between the supplier and each retailer, if the supplier wants to set a different price per retailer.

Figure 6 illustrates a distributed publish–subscribe architecture that has been integrated on top of the Hyperledger Fabric. Publishers generate data, and distributed brokers integrated into the Blockchain platform will receive the data from the publishers, verify, and validate the data content, and coordinate data communications between publishers and subscribers. Subscribers who have already registered for specific data should then receive the verified data via the brokers. Publishers and subscribers usually belong to different organizations and do not know each other. This situation is similar to a SC in which IoT devices are connected to the different parts of particular organizations. These participants are interfaced with their own brokers to maintain a consistent state through the consensus protocol [72]. A hybrid storage model that includes both on-chain and off-chain storage is used. IPFS can be used for off-chain data storage, as more data are sent on the Blockchain and can be extended with additional storage using cloud facilities [73,74].

#### 4.2.4. Application Layer

The system’s functionalities in the application layer are exposed to the end-users, who can collaborate with the system using write operations to store data, queries to retrieve data, remote visualizations, data analytics, image processing, and ML. For example, the information extracted from the knowledge layer is processed and visualized over the fisher APP, enabling fishers to take Blockchain-verified photos to document their catch. The photos are then uploaded to the Blockchain along with the corresponding timestamp and the digital certificates. Origin proofs and real-time traceability are possible using this method. Stakeholders, including authorities, business partners, and consumers, can also use traceability software to track product history from capture sites to the consumer.

Another example is consumer APPs that can be used to verify the purchased products in terms of provenance, traceability, and quality and provide nutrition information to the final consumers. Integrating image processing and ML with fish SC systems can assist in detecting fish quality and safety, especially with the recent advances in handheld inspection devices (Section 5.2). Blockchain’s main benefit is integrating all SC parties into a single secure network; however, many challenges may arise when implementing BC in SC systems, including (1) organizational challenges related to standardization and organizational readiness, (2) technical challenges related to the technology’s immaturity and cost, and (3) operational challenges related to SC operations and logistics management [75], which are out of the scope of this work.

## 5. Machine Learning Integration with Blockchain IoT-Enabled Supply Chain

Blockchain IoT-enabled fish SC systems are expected to collect abundant amounts of data regarding fish welfare, water quality, genetic information, feed, processing, and distribution, which can be used to advance SC management systems toward greater sustainability and transparency. ML can be used to automate data analysis and business decisions by categorizing, standardizing, aggregating, annotating, and transforming the large and unstructured data coming from different areas of the SC [76]. Trained ML models can then be used to construct applications, solutions, or tools for efficient big data management and sustainable Blockchain-based SC decision making, reducing human intervention and overcoming decision-making weaknesses of traditional methods [77]. A ML model’s effectiveness is largely determined by dataset availability and the algorithm, which can be chosen from well-known ML categories including supervised, unsupervised, semi-supervised, reinforcement, and federated learning. Blockchain combined with IoT technologies including WSNs, handheld devices, and automated identification systems support high-quality and reliable data collection.

### 5.1. Related Work

Several applications that integrate Blockchain and IoT technologies with ML have been discussed in the literature, including the prevention of IUU activities. For example, OpenSC developed a Blockchain-based SC solution integrated with ML. A GPS dataset was collected from geo-locations, stored on the Blockchain-based traceability system, and used to develop ML models. This system ensures that fish are caught in legal fishing zones and are sustainably and ethically produced [78]. The authors of [19] discussed integrating ML with Blockchain-based SC systems to obtain insights from data analysis on optimizing SC services, including reducing transport costs and optimizing shipment scheduling. A case study on using an ML algorithm for logistics and shipment purposes was presented by Maersk and IBM [79]. The learning process allows the system to determine better shipping routes and safe zone decisions. The authors were able to predict the shipping routes and times by training the ML algorithm on historical shipment data and by incorporating external information sources related to scheduling reliability. The authors of [80] proposed a Blockchain-based food traceability system. The system predicted the estimated food product’s expiration date based on environmental information collected via IoT sensors, such as temperature, humidity, and oxygen level. The authors of [81] discussed a Blockchain-based drug SC system using ML to recommend the most effective and safest drugs to consumers. The authors of [82] generalized the purpose of using ML in Blockchain agri-food SC systems to make informed decisions on smart transportation, logistics, and quality control. The authors proposed a grape wine SC integrated with Blockchain.

Table 4 summarizes both industrial and academic SC solutions using ML and Blockchain integrated methods. Table 5 depicts the key roles of ML integrated with Blockchain IoT-enabled fish/food SC including fishing, manufacturing, shipping and transportation, logistics, customer service, healthcare, quality control, assets maintenance and replacement, labor, pricing prediction, fraud detection, and contamination inspection.

Several aspects need further study and in-depth investigation to improve quality and safety compliance, to facilitate fish/food global trade, and to secure SC systems against IUU activities. It is essential to secure SC systems to avoid fish fraud through fraud detection and prevention methods. Fraud prevention means stopping possible fraudulent occurrences [3] using Blockchain by preventing product manipulation, while ML can be used to develop efficient detection models to assess fish quality and to identify possible fraudulent occurrences [44]. These models will be trained and tested using large datasets consisting of, for example, video records and hyperspectral images, collected at checkpoints along the SC using physical and optical onsite sensor devices and cameras [85]. This way, Blockchain eliminates some pre-processing overhead while ML provides efficient data-driven decisions.

The authors of [86] used a deep learning technique, convolutional neural network, for automatic fish weight estimation. The dataset was a collection of harvested fish images taken with a video camera and their corresponding weight values. The authors of [87] used the same technique to propose an automatic fish species classification model using thousands of fish images captured in different environments. These ML techniques can be used to differentiate fraudulent occurrences in fish products. Other studies focused on raw fish species testing and DNA analysis using efficient ML techniques toward developing in-field usable handheld detection devices with AI capabilities. The devices allow stakeholders to avoid time-consuming laboratory tests and data analysis by detecting fish origin, monitoring the freshness, tracking changes in freshness, identifying seafood species based on their unique features, assessing quality, and notifying the system of any quality or health issues [88,89]. For example, the authors of [44] targeted the detection of possible fraudulent activity using hyperspectral imaging coupled with selected ML classification techniques, such as decision trees, discriminant analysis, naive Bayes, support vector machines, k-nearest neighbor, and ensemble methods to classify species, inspect for freshness, and detect fish fillet substitution and mislabeling.

ML and SC system integration has recently drawn a large amount of attention from both the industry and academia. Few studies have integrated ML with Blockchain-based SC systems [90]. Learning-based analyses of Blockchain-based systems are rare, despite the existence of big data, which can be used to build efficient prediction models. Some researchers have used supervised learning and deep learning techniques for fish SC; however, only a few have explored the potential of other ML techniques, especially reinforcement learning and unsupervised learning. Reinforcement learning and unsupervised ML can ultimately perform better in cases such as fish market prediction, since historical data cannot reflect the dynamic market due to a lack of data samples for fish species of certain types.

### 5.2. Commercial Use Case

Existing tracking technologies for seafood products, such as RFID, barcodes, WSNs, and checklists, lack the ability to authenticate key attributes of the seafood, such as species, freshness, and provenance. SafetySpect Inc. is developing a multimode spectroscopic handheld device with AI models developed through ML that can be used to facilitate onsite spot-check measurements of fish species identifiers and that assess fish quality and nutrient content (Figure 7). The three modes are fluorescence spectroscopy, visible near-infrared spectroscopy and short-wave infrared spectroscopy. Fluorescence spectroscopy analyzes the different components of a fish tissue, such as collagen, amino acids, NADH, FAD, and proteins. Visible near-infrared and short-wave infrared reflectance spectroscopy analyze the physical properties of the fish, such as hemoglobin, water, lipids, proteins, chemical signatures such as CH/OH, and tissue-scattering properties. The AI models are based on data fusion of the three modes of spectroscopy. Separate models are implemented to identify species to assess freshness and other quality factors. For 37 species, we achieved an accuracy of greater than 95%. By comparison, DNA analysis has an accuracy of 98%.

Users should be able to compare the product species label with the actual species, as well as the geographic origin and other important product attributes related to fish fraud. Blockchain-based SC system, coupled with such spectroscopic technology implemented on IoT-enabled devices, can ensure trust in the data generated throughout the entire SC system. The commercial application of the proposed Blockchain IoT-enabled fish SC framework can be designed and implemented using platforms such as Hyperledger Fabric and modern cloud-based solutions with the potential to add quality and freshness assessments for the fish/seafood species by incorporating new handheld IoT-enabled devices for real-time measurements. The spectroscopic technologies implemented with newly developed ML classification techniques in the form of handheld inspection devices have the potential to optically detect the established chemical signatures of the seafood species and quality. The collected spectroscopic data are used to generate simple, readable reports as well as data that are accessible to system parties and that can be incorporated in BC.

## 6. Conclusions

SC is a promising field for the application of Blockchain to track and trace fish products and to verify their authenticity throughout the chain. Integrating IoT and Blockchain in an SC system allows for better trackability and traceability for the product during its movement throughout the chain. Fish and seafood are among the most traded food products; however, they are susceptible to illegal, unreported, and unregulated activities. We reviewed the industry solutions and research works that have utilized Blockchain in implementing smart fish SC systems. We have also proposed a Blockchain IoT-enabled fish SC framework. Layered architecture is used for the proposed framework design. In this system, fish can be processed, packaged, and shipped several times during its journey, while Blockchain records all data transactions in an immutable ledger shared between stakeholders. Blockchain assures system users that the relayed information is unalterable, trusted, and accurate. We also established the roles of ML integration with Blockchain IoT-enabled SC systems, focusing the discussion on preventing and detecting fish fraud.

Several technical issues must be addressed if a fully traceable, trustworthy SC system that operates with Blockchain technology is to be realized. Interoperability, scalability, privacy, and latency are among the issues that will be the focus of upcoming research necessary for successful implementation. Other barriers to Blockchain adoption in different SC applications include cost, legal concerns, and organizational resistance, which can be managed by standardization, compatibility, and adequate infrastructure.

Our future work includes implementing the proposed framework using the Hyperledger Fabric platform and focusing on the design of business logic and smart contracts, which organize stakeholder business practices and transaction automation. Future work may also include analyzing the proposed framework with respect to the quality model presented in this paper.

## Figures and Tables

**Figure 1 sensors-23-05136-f001:**
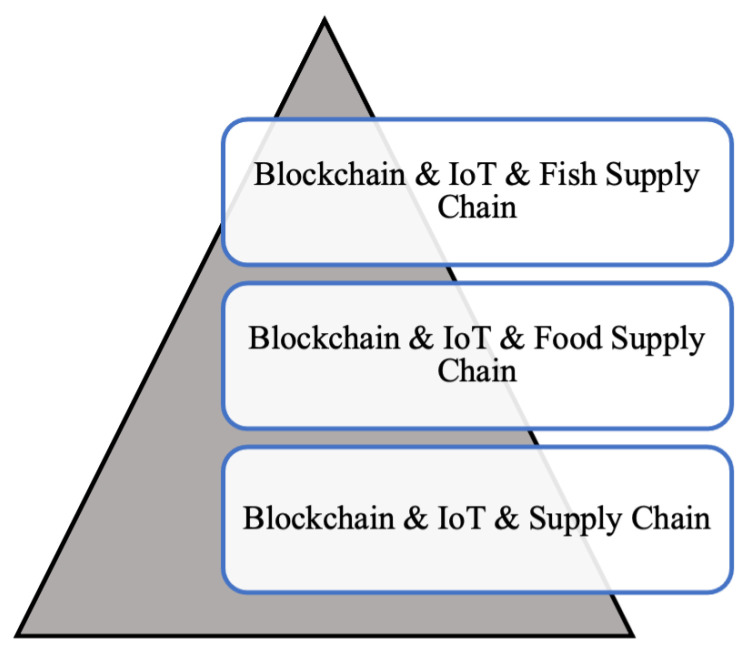
IoT, Blockchain, and SC integration levels.

**Figure 2 sensors-23-05136-f002:**
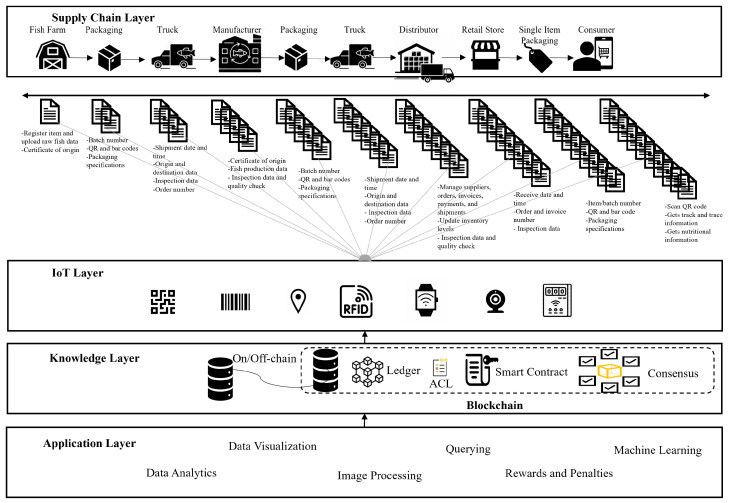
Layeredarchitecture of the proposed Blockchain IoT-enabled SC system.

**Figure 3 sensors-23-05136-f003:**
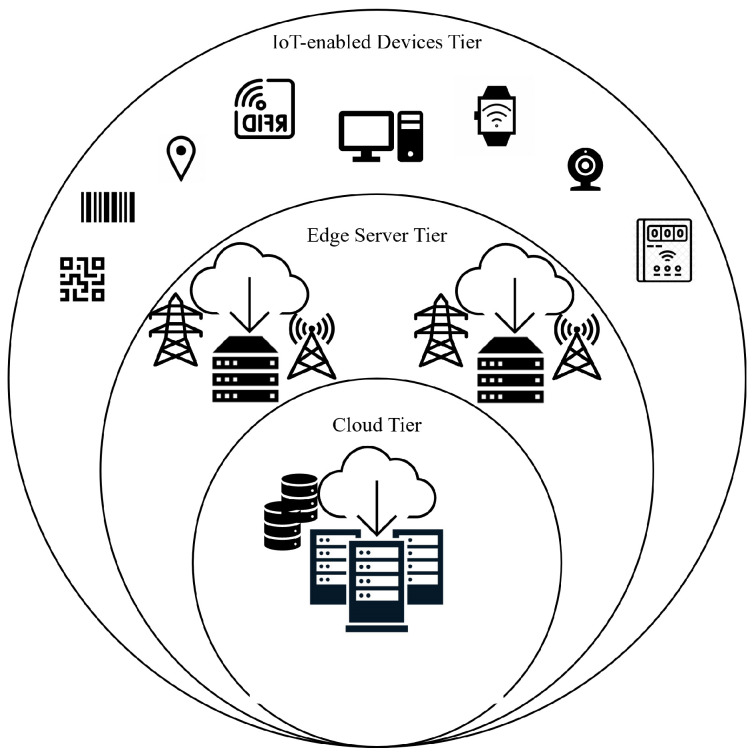
Three-tiered IoT layer network architecture.

**Figure 4 sensors-23-05136-f004:**
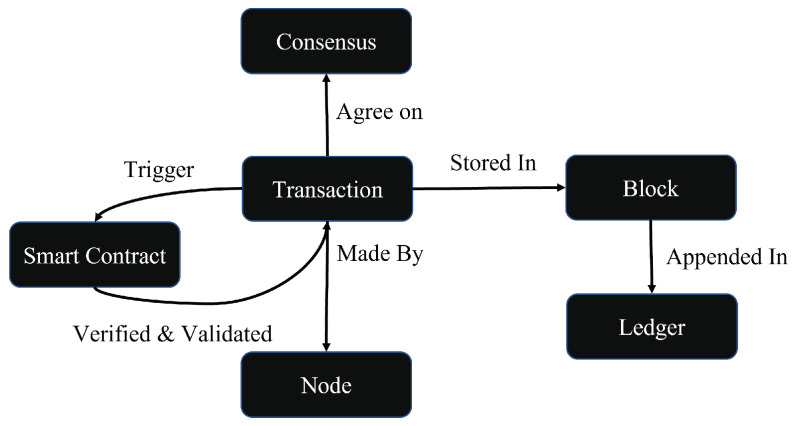
Essential components of Blockchain ontology.

**Figure 5 sensors-23-05136-f005:**
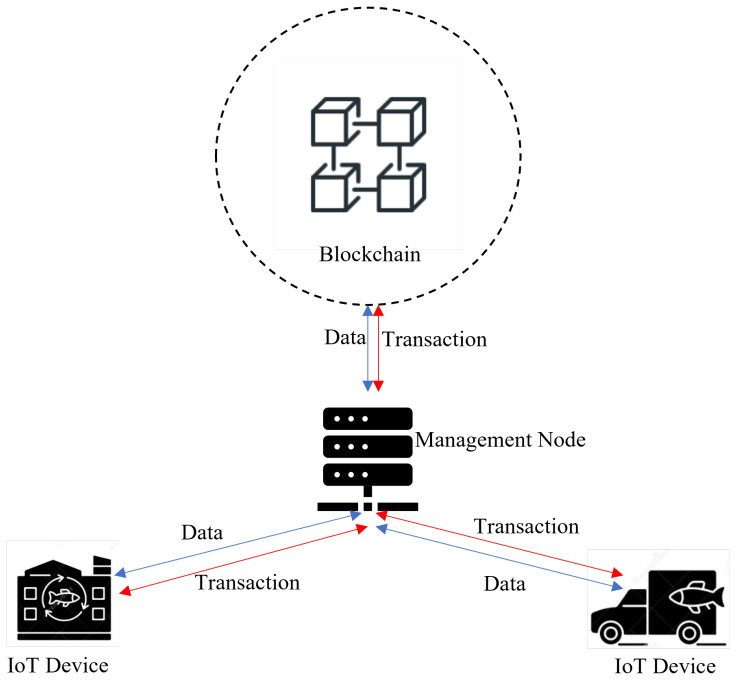
Proposed system design pattern.

**Figure 6 sensors-23-05136-f006:**
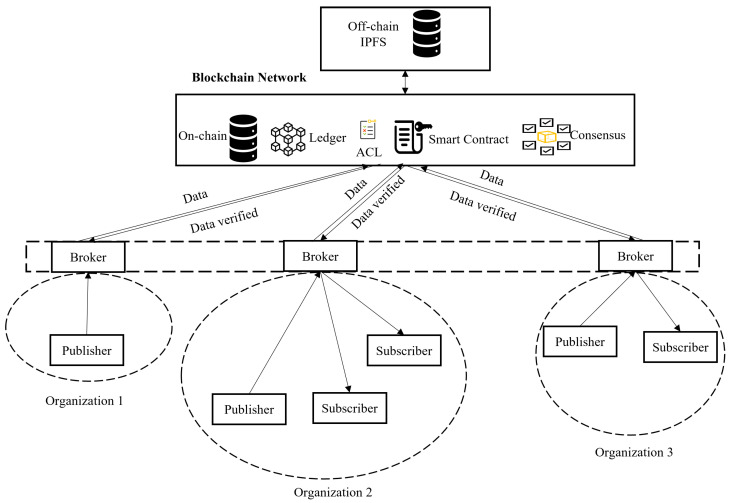
Blockchain-based publish–subscribe architecture.

**Figure 7 sensors-23-05136-f007:**
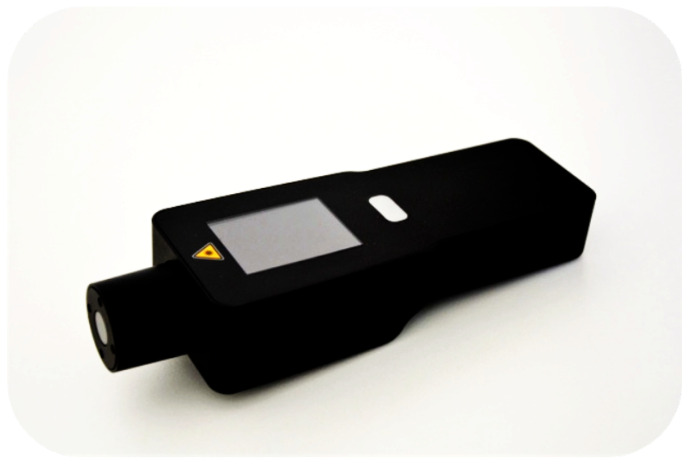
SafetySpect’s QAT handheld device.

**Table 1 sensors-23-05136-t001:** Core architectural requirements of Blockchain-based SC systems.

Quality Attribute	Design Considerations
Traceability	Track and trace the moving product throughout the SC during its lifetime, influenced by a variety of factors such as: (1) smart contracts, (2) business logic design, (3) transactional rate and processing speed, and (4) data structure.
Performance	Blockchain-based system performance is affected by resource availability in terms of storage and processing. Factors to consider: (1) Blockchain platform ( private or consortium is preferred), (2) data structure and block size, (3) transactional rate and processing speed, and (4) choice of consensus protocol.
Security	Consider possible malicious activities and security risks, especially with the integration of IoT-enabled devices. Factors to consider: (1) choice of a consensus protocol, (2) access control lists (ACLs) and trust establishment.
Privacy	Ensure the confidentiality of stakeholders’ sensitive information. Factors to consider: (1) user identity and product authentication and (2) Blockchain platform (private is preferred).
Interoperability	Ability to communicate and access information across various Blockchain systems by standardizing Blockchain implementation and ensuring regulation compliance especially with IoT integration. Factors to consider: (1) Blockchain-based layered architecture, (2) standardized language, and (3) on-chain computation.
Scalability	Adapt to a growing number of users and IoT-enabled devices, data ledger size, transactional rate and processing speed, and data transmission latency. Factors to consider: (1) Blockchain platform, such as private or consortium, (2) network size, (3) block size and transaction size, (4) off-chain and cloud storage, (5) choice of consensus protocol, and (6) sharing.
Latency	Reduce the time taken to transfer the data. Latency is affected by transactional rate, processing speed, and security check time. Factors to consider: (1) off-chain and cloud storage, and (2) Blockchain platform (private is preferred to avoid long commitment times compared to public).
Integrity	Manage an immutable and permanent data ledger that cannot be altered or deleted and maintain identical copies across the nodes. Factors to consider: (1) off-chain storage and (2) choice of consensus protocol.
Usability	Ease of use and user ability to achieve the desired outcome and to obtain access to data. Factors to consider: (1) user-friendly application programming interface (API) and (2) multi-factor authentication.

**Table 2 sensors-23-05136-t002:** Use case sample.

Use Case	
Use Case Name	Automatically reading observations from an IoT-enabled sensor.
Primary Actor	IoT-enabled sensor.
Description	Reading an observation from an IoT-enabled sensor.
Input Data	Data readings from an IoT-enabled sensor such as temperature, humidity, geo-location.
Pre-Condition	The IoT-enabled sensor is connected to the network, either wired or wireless. The IoT-enabled sensor is active and able to detect and measure physical surroundings automatically and promptly.
Post-Condition	A new observation is sent over the network in the form of a data transaction. The Blockchain system validates and verifies the transaction to be combined with other transactions in a new block and added to the chain.
Main Flow	The IoT-enabled sensor observes a new reading automatically and promptly. Data observation uses a procedure with input and output. The IoT-enabled sensor generates a write transaction for the observation. The transaction is sent to be verified and validated and then added to the database.
Alternative Flow	None.

**Table 3 sensors-23-05136-t003:** Abusecase sample.

Misuse Case	
Misuse Case Name	Fraud, product data tampering.
Category	Security Attacks.
Goal	Adulteration by tampering with the data stored on the product tag and code, such as RFID tag and QR code.
Primary Actor	System, fraudulent stakeholder.
Description	Fraud through false tag data.
Input Data	Data stored on RFID tag, QR code, and the system.
Pre-Condition	Each fish product is assigned a unique RFID identifier and QR code. The fish product data stored on the RFID tag or QR code consists of: (1) production origin, area, state, and country; (2) product weight at the time of packaging, which may have several packaging stages; (3) packaging time and date; and (4) additional product information such as product grade, product quality assessment features, and expiration date. The same data on the RFID tag and QR code are stored on the Blockchain. The RFID tag and QR code are attached to the fish product or container.
Attack Flow	Fraudulent participant may tamper with the data from the product’s RFID tag or QR code, or false information is injected.
Post-Condition	The system has false information about the product.
Detection Flow	Trace-back is performed to detect any possible fraud. The likelihood of product fraud is inferred if the information on the Blockchain and RFID tag or QR code does not match. No product fraud is inferred if the information on the Blockchain and RFID tags matches.

**Table 4 sensors-23-05136-t004:** Review of Blockchain-based SC systems integrated with ML (N/A means not available).

Ref/Type	Machine Learning Technique	Product Type	Study Objective	Dataset
[83]/Industry	N/A	Fish	Verifying the vessel’s GPS locations to determine if it is within legal fishing zones	Dataset of satellite imagery, live video monitoring
[84]/Industry	N/A	Shipment and logistics	Enhancing supply chain services such as transport cost reduction and shipment schedule optimization	Geo-location data collected through GPS sensors
[80]/Academic	Bayesian Regression and Random Forest	Food	Predicting the product’s estimated expiration date	Real-time dataset that was collected from temperature and humidity sensors
[81]/Academic	LightGBM	Drug	Recommending drugs to system users	Drug reviews dataset provided by the UCI
[82]/Academic	Dummy Variable Regression	Grape wine	Informed decisions for smart transportation and logistics and quality control	Survey questionnaire for collecting data from the actors

**Table 5 sensors-23-05136-t005:** Roles of ML integrated with Blockchain IoT-enabled SC systems.

Integration Roles	Description
Fishing	Ensuring safe and legal fishing zones. Reducing the time needed for video review and lowering the cost of electronic monitoring. Visual tracking for automatic fish counting. Decision making in the management of fishing activities such as season fishing dates and allowable catch tonnage based on demand prediction.
Manufacturing	Developing ML models to assist manufacturers in factories to make informed decisions on the quality of the fish and to decide the best preliminary processing or refining steps.
Shipping and Transportation	Ensuring optimized shipping routes and excluding shipment danger zones. Predicting shipping time and routes (dynamic routing). Identifying the direction of the tagged fish. Predicting transport time and routes to shorten the distribution time to guarantee fish freshness.
Logistics	Dynamic inventory management to predict demand/sales of fresh fish in the near future, allowing stock to be purchased on time.
Customer Service	Providing real-time recommendations to consumers about products using models developed with datasets collected from consumer reviews.
Health	Sustainable food fishing for the health of ecosystems (marine). Predicting water freshness for the health of the fish product. Estimating product expiry dates for consumer health.
Quality	Finding fish quality patterns for each supplier and automating track-and-trace reporting. Automating quality inspection throughout the supply chain phases, for example, determining damage in shipping containers, classifying it by damage type and time, and recommending the best corrective action to repair the assets.
Assets Maintenance and Replacement	Finding patterns in hardware asset usage to establish the factors that most influence devices and machinery performance.
Labor and stakeholders assessment	Performance assessment, labor status prediction.
Pricing	Finding optimal pricing based on seasonal demand, market feedback, the grade/species, and prevailing wholesale market price.
Fraud Detection	Predicting the occurrence of fish fraud and other malicious threats and identifying the factors leading to fraudulent activities.
Contamination	Predicting the hazard type. Detecting fish contamination with the new specialized handheld contamination and inspection devices.

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
