# Peer review of "Toward an Intelligent Blockchain IoT-Enabled Fish Supply Chain: A Review and Conceptual Framework"

_sensors, 2023, doi:10.3390/s23115136_

Round 1
Reviewer 1 Report
The manuscript is well formatted and the topic is interesting. The only issue I would recommend to further elaborate is related to the sources. It would be beneficial to add more sources published in 2023 since only 3 such sources are currently used. Also the influence of covid is only briefly mentioned in the beginning. I would recommend to expand the information provided on this topic in conclusions by adding those more recent sources of information that already document some post-pandemic changes in supply chains that can be interesting to support your arguments.
Author Response
Thank you for your time reviewing our paper.
The discussion on COVID influence on supply chain market is extended (section 2, page 3). We add 8 additional references: 3 published in 2023, 2 published in 2022, 2 published in 2021, and 1 published in 2020.
Best,
Reviewer 2 Report
This paper is interesting and well written. But it requires a major review to address my following comments.
1) I suggest to shorten Section 2 and integrate it in the introduction Section.
2) Moreover, I recommend to add reference papers that discuss the blockchain and the traceability in supply chain of other industrial fields.
For instance:
a) Pharmaceutical:
Chiacchio, F., D’Urso, D., Oliveri, L. M., Spitaleri, A., Spampinato, C., & Giordano, D. A Non-Fungible Token Solution for the Track and Trace of Pharmaceutical Supply Chain.
b) Wine:
Brookbanks, M., & Parry, G. (2022). The impact of a blockchain platform on trust in established relationships: a case study of wine supply chains. Supply Chain Management: An International Journal.
c) Automotive:
Reddy, K. R. K., Gunasekaran, A., Kalpana, P., Sreedharan, V. R., & Kumar, S. A. (2021). Developing a blockchain framework for the automotive supply chain: A systematic review. Computers & Industrial Engineering, 157, 107334.
3) I suggest to shorten Section 3 and move it in what is now Section 6. Much information are well-known and the technical part (eg. miners) is not stricly related to the topic.
4) I suggest to collapse Section 4 and 5 to become Section 2: Literature Review
5) Section 6 could inherit some info of the previous Section 3. Thus, traceability can be discussed as one of the main attribute of this technology.
6) I don't see the point of discussing the SW Traceability in such manuscript. I don't see the reason to address the concept of Traceability from several angles.
What it matters is to explain the benefits of blockchain in supply chain traceability.
Please simplify the Section 6.
--
Data Storage and Computation, and Consensus Protocols are minor and too technical topics.
7) Requirements and Architectural Framework can become a unique section: Proposed Intelligent Blockchain Solution
8) Section 9 and 10 can be merged.
To be honest the section 9 should be expanded to discuss what use case have been addressed by the authors with the proposed prototype and how they think - or what is needed - for this solution to be commercial.
In this way, the role of machine learning can be used as a leverage for further potential benefits.
Otherwise, I think that this topic is yet detached from the main research proposed by authors.
9) The conclusions should be improved to discuss also the limitations and the current barriers.
Author Response
Thank you for your time reviewing our paper. Please note that that we consider reorgnizing the manuscript according to your comments. Our response is attached.

Reviewer 3 Report
see the attached, please.

Author Response
Thank you for your time reviewing our paper. Appreciate your support recommending our manuscript for submission.
Best,